



# Evaluation and optimisation of the I/O scalability for the next generation of Earth system models: IFS CY43R3 and XIOS 2.0 integration as a case study

Xavier Yepes-Arbós[1], Gijs van den Oord[2], Mario C. Acosta[1], and Glenn D. Carver[3]

[1]Barcelona Supercomputing Center - Centro Nacional de Supercomputación (BSC-CNS), Barcelona, Spain
[2]Netherlands eScience Center (NLeSC), Amsterdam, The Netherlands
[3]European Centre for Medium-Range Weather Forecasts (ECMWF), Reading, United Kingdom

**Correspondence:** Xavier Yepes-Arbós (xavier.yepes@bsc.es)

**Abstract.**

Earth system models have considerably increased their spatial resolution to solve more complex problems and achieve more realistic solutions. However, this generates an enormous amount of model data which requires proper management. Some Earth system models use inefficient sequential Input/Output (I/O) schemes that do not scale well when many parallel resources are

used. In order to address this issue, the most commonly adopted approach is to use scalable parallel I/O solutions that offer both computational performance and efficiency.

In this paper we analyse the I/O process of the European Centre for Medium-Range Weather Forecasts (ECMWF) operational Integrated Forecasting System (IFS) CY43R3. IFS can use two different output schemes: a parallel I/O server developed by MeteoFrance used operationally, and an obsolescent sequential I/O scheme. The latter is the only scheme that is being exposed

by the OpenIFS variant of IFS. 'Downstream' Earth system models that have adopted older versions of an IFS derivative as a component –such as the EC-Earth 3 climate model– also face a bottleneck due to the limited I/O capabilities and performance of the sequential output scheme. Moreover it is often desirable to produce gridpoint-space Network Common Data Format (NetCDF) files instead of the IFS native spectral and gridpoint output fields in General Regularly-distributed Information in Binary form (GRIB) format, which requires the development of model-specific post-processing tools.

We present the integration of the XML Input/Output Server (XIOS) 2.0 into IFS CY43R3. XIOS is an asynchronous Message Passing Interface (MPI) I/O server that offers features especially targeted at climate models: NetCDF format output files, inline diagnostics, regridding, and when properly configured, the capability to produce CMOR-compliant data. We therefore expect our work to reduce the computational cost of data-intensive (high-resolution) climate runs, thereby shortening the critical path of EC-Earth 4 experiments.

The performance evaluation suggests that the use of XIOS 2.0 in IFS CY43R3 to output data achieves an adequate performance as well outperforming the sequential I/O scheme. Furthermore, when we also take into account the post-processing task, needed to convert GRIB files to NetCDF files, and also transform IFS spectral output fields to gridpoint space, our integration not only surpasses the sequential output scheme but also the IFS I/O server.



## 1 Introduction

Over the years, the computing power of High-Performance Computing (HPC) has grown exponentially (Poyraz et al., 2014). Earth System Modelling (ESM) has employed it to considerably improve the accuracy of weather forecasts and the fidelity of climate models (Yashiro et al., 2016) by increasing both the computational complexity of the models and their spatio-temporal resolution. The added complexity has accommodated improved physics parameterisations and the introduction of new components simulating the interaction with secondary processes (biochemistry, ice-sheet, etc.). Increasing the model resolution has enabled higher accuracy of the underlying fluid dynamics, yielding for example better representation of convection, cloud processes and turbulent fluxes, leading to a more faithful simulation of key phenomena such as the Gulf Stream (Chassignet and Marshall, 2008), tropical cyclones (Zhao et al., 2009), and the global water cycle (Demory et al., 2013). As a result, high-resolution climate models have been shown to improve seasonal predictability (Prodhomme et al., 2016), reduce persistent biases in for example the Tropical Pacific Ocean (Roberts et al., 2009), and enable systematic studies of regional climate that account for the transition of small-scale phenomena to large-scale weather, such as the development of Atlantic hurricanes into storms in western Europe (Haarsma et al., 2013).

The growing burden of model output is one of the key computing aspects at higher model resolutions, especially for the climate community. One of the immediate challenges is to efficiently write the larger time slices to disk during the model run, preferably without halting the entire execution of the parallel code. A second potential problem is the post-processing of model output data: regridding (or spectral transformations), data reduction through time or spatial averages, computing derived diagnostics, etc. These tasks can prove hard to parallelise efficiently and impose a heavy burden on the storage system. Finally, transferring and analysing the resulting data becomes a more demanding process, which are issues that will not be addressed in this paper.

Although the ESM community has made considerable efforts in using HPC techniques to improve the hardware utilisation and scaling of algorithms (Jackson et al., 2011), the Input/Output (I/O) performance aspect has not received as much attention because it was not deemed critical enough (except in operational weather forecasting). Improved I/O efficiency is however becoming a necessary ingredient to sustain the throughput of next-generation Earth system models, with their increasing resolution, output frequency and growing number of diagnostics. With the exascale era approaching rapidly, an inefficient output scheme that blocks the model time stepping and fails to utilise the network bandwidth to the parallel file system (Liu et al., 2013) will become a bottleneck, and may prevent the model from taking advantage of this enormous compute capability.

Among the codes that may run into this conundrum are Earth system models that build upon the Integrated Forecasting System (IFS). IFS (Barros et al., 1995) is a global data assimilation and forecasting system developed by the European Centre for Medium-Range Weather Forecasts (ECMWF) and used by several institutions in Europe. IFS has two different output schemes: an efficient parallel I/O server using dedicated resources, and an obsolete sequential output scheme which gathers all data and writes via the single master process. While ECMWF uses the IFS I/O server for its operational forecasts and research, partner institutions using the OpenIFS derivative are bound to employ the sequential output scheme, as the parallel I/O server code of IFS is not provided with OpenIFS. We describe IFS and OpenIFS in more detail in Sect. A1.





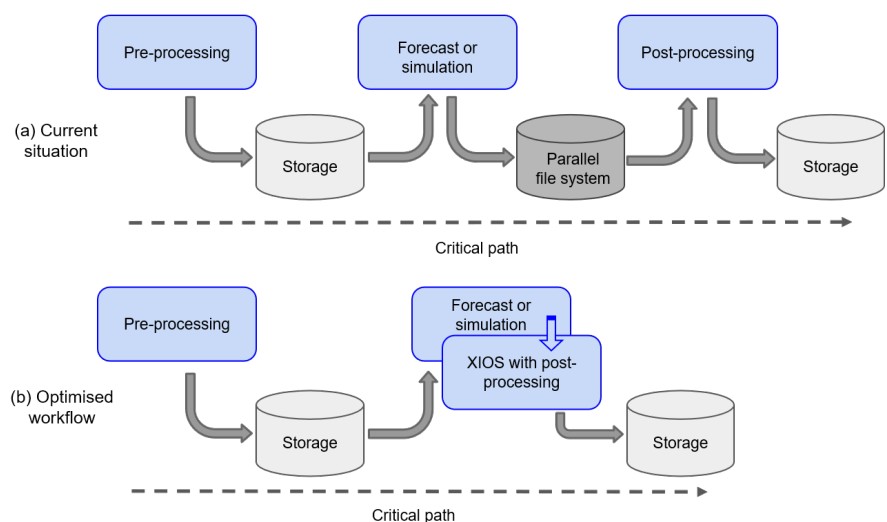

**Figure 1.** Schematic showing the typical key components, involving pre- and post-processing of the critical path of a forecast or climate simulation. The arrows indicate the flow of the input and output. When using XIOS to perform the post-processing, the critical path is more efficient as post-processing no longer requires input/output from file system storage and post-processing happens concurrently to the simulation.

One Earth system model that comes to mind is EC-Earth 3 (Hazeleger et al., 2010), a global circulation model (GCM) that couples IFS (based on the ECMWF IFS CY36R4 operational code) to the Nucleus for European Modelling of the Ocean (NEMO) 3.6 and other Earth system components, using the Ocean Atmosphere Sea Ice Soil 3 Model Coupling Toolkit (OASIS3-MCT). Climate models such as EC-Earth are a good example to demonstrate why an efficient I/O scheme for

the atmospheric sub-model is needed. The EC-Earth 3 experiments for the PRIMAVERA project (http://proj.badc.rl.ac.uk/primavera) (Haarsma et al., 2020), consisting of a 100 years projection, generated a total amount of 244 TB of data, with I/O accounting for 30% of the total execution time. For a typical EC-Earth simulation requiring over 2 million core hours, this represents a significant waste of processor time caused by the serial I/O blocking the parallel model execution. This time does not include the post-processing stage, which adds a significant additional computation time; EC-Earth atmospheric output needs

to be converted from General Regularly-distributed Information in Binary form (GRIB) files output to Network Common Data Format (NetCDF) files. Whereas GRIB, the de-facto standard in Numerical Weather Prediction (NWP), is a highly compressed data format designed to offer high I/O performance in the time-critical operational forecast, NetCDF is the one accepted format for coordinated climate model intercomparison projects such as CMIP6 (Eyring et al., 2016). Furthermore, the raw EC-Earth output requires a transformation of spectral fields to gridpoint space and production of additional derived diagnostics to comply

with the CMIP6 data request. This workflow –including pre-processing of input such as forcing fields or initial conditions– has been depicted in the workflow of Fig. 1*a* and we refer to it as the *critical path*, the series of inter-dependent tasks that must be executed in sequence.



The work we present in this manuscript aims to shorten the critical path for Earth system models that incorporate IFS or OpenIFS by optimising the I/O scheme and providing the foundations to absorb the post-processing into the model execution. We start in Sect. 2 with an extensive motivation of this work, and discuss related efforts. An analysis of the two IFS output schemes is given in Sect. 3. Section 4 explains the development done for the IFS CY43R3 and XIOS 2.0 integration, as well as the optimisation techniques applied. The computational performance of the integration is evaluated in Sect. 5, followed by the model output validation in Sect. 6. Finally, we summarise the work and present conclusions in Sect. 7.

## 2 Motivation and related work

Where IFS (and OpenIFS) offers the performance, stability and numerical accuracy to be run at seasonal, decadal and climate timescales, it does not offer the flexibility to configure its output to comply with widely accepted standards in the climate community. Furthermore, the old sequential IFS I/O scheme that EC-Earth 3 still uses incurs a heavy performance penalty for higher-resolution model versions and output-intensive experiments. This scheme is certainly a bottleneck which prevents the EC-Earth community to push the resolution of IFS to ultra-small eddy-resolving scales. We intend to resolve these issues by providing a concurrently executing parallel XIOS I/O server to the IFS output production module. We foresee the following benefits of this integration:

1. Improve scalability of IFS derivatives that have no access to the ECMWF operational parallel I/O server. XIOS is capable of writing output in parallel and as such provides a better utilisation of the storage system bandwidth. Furthermore, XIOS output servers can asynchronously perform this task without blocking the IFS time stepping.

2. Providing a mechanism to produce NetCDF files from IFS, optionally with user-provided metadata. XIOS is also capable of post-processing the IFS fields, and computing additional diagnostics, such as time-means, by its server processes. Furthermore we transform all data to gridpoint space, eliminating the requisite to do this afterwards, as depicted in the workflow of Fig. 1*b*. When configured appropriately, XIOS can produce all CMIP-compliant data for IFS. Note that inline post-processing, which consists of concurrently running the model time stepping with the post-processing, also often reduces the output significantly, thereby contributing to the I/O performance as well.

3. Providing a more integrated output configuration strategy for coupled models which employ XIOS already, such as EC-Earth. Since XIOS already handles output from NEMO, a more uniform configuration across ocean and atmosphere can be pursued for EC-Earth. Also, derived fields combining output from multiple sub-models will be feasible.

4. Alleviating the engineering effort to secure a high-performance I/O strategy from the climate science community towards the developers of XIOS, thereby leveraging the expertise of computer scientists and research software engineers, and vice versa, increasing the impact of the XIOS initiative throughout the European climate and weather modelling landscape.

We stress that the idea of deploying parallel I/O with dedicated servers and inline diagnostics for Earth system models has a long history, and almost all state-of-the-art climate models adopt these techniques in some way. Parallel I/O is usually achieved





by inserting new layers in the I/O software stack between the application and the parallel file system: a high-level I/O library and the required I/O middleware layers. For the latter, the most commonly adopted library is the MPI-IO (Message Passing Interface Forum, 2003), whereas for the high-level layer one usually encounters either HDF5 (Folk et al., 2011), PnetCDF (Li et al., 2003) or NetCDF4 (https://www.unidata.ucar.edu/software/netcdf/). Other, less widespread I/O libraries include PIO
(Dennis et al., 2012) and Gtool5 (Ishiwatari et al., 2012).

There is a particular class of parallel I/O tooling that uses dedicated computing resources to exclusively perform I/O, known as I/O servers. They are separately executing processes responsible for writing data into the storage system in order to hide the disk latency from the model processes, and use the network bandwidth as efficiently as possible e.g. by aggregating writes. Some of the I/O servers available in the literature are: ADIOS (Jin et al., 2008), CDI-pio (https://code.mpimet.mpg.de/projects/
cdi), CFIO (Huang et al., 2014), Damaris (Dorier et al., 2012) and XIOS (Joussaume et al., 2012).

There have been efforts to improve the tools used to analyse the I/O performance of parallel applications by producing lighter traces in size with a low runtime overhead (Vijayakumar et al., 2009)(Uselton et al., 2010). Other studies adopt novel techniques to improve I/O performance, such as the ones from Gao et al. (2009) and Nisar et al. (2008). In addition, Poyraz et al. (2014) gives an overview of different optimisations such as file system striping, data aggregation, interleaving of data, collective MPI-
IO and data staging. Liu et al. (2013), Zou et al. (2014) and Tseng and Ding (2008) have reported case studies involving the climate models GEOS-5, CAM and GRAPES respectively. Kern and Jöckel (2016) integrates a diagnostic interface into the ICON model to apply post-processing operations during runtime to achieve output reduction.

We have selected the XML Input/Output Server (XIOS) as our I/O library of choice because it provides the flexibility and performance needed to serve the climate community, and because it is widely used within the European ESM groups. The
abundant technical expertise of XIOS increases the chances of our work to be adopted by other parties, such as the EC-Earth consortium. We describe XIOS in full detail in Sect. A2.

# 3 Analysis of the IFS CY43R3 I/O schemes

## 3.1 Communication and I/O strategy

The old sequential output scheme of IFS is characterised by performing serial writing. First, all processes independently
perform an internal post-processing with the IFS FullPos diagnostic output package (ECMWF, 2017). This component is mainly dedicated to vertical interpolation, e.g. to user-defined pressure levels. Then, 2D and 3D fields are gathered by the master Message Passing Interface (MPI) process, which encodes the data in GRIB format. Finally, this MPI task sequentially writes data onto the storage system. At high resolutions one expects excessive memory usage at the I/O node after the gathering and long waiting times for the I/O process to flush the field buffers.
The IFS I/O server in the ECMWF operational IFS uses the concept of dedicated server processes to execute the model I/O. IFS sends data to these servers using asynchronous MPI communications to achieve high throughput and faster runtime. However, unlike XIOS, this output server lacks the flexibility to define derived diagnostics or perform time averaging, nor does it support writing anything other than GRIB files.





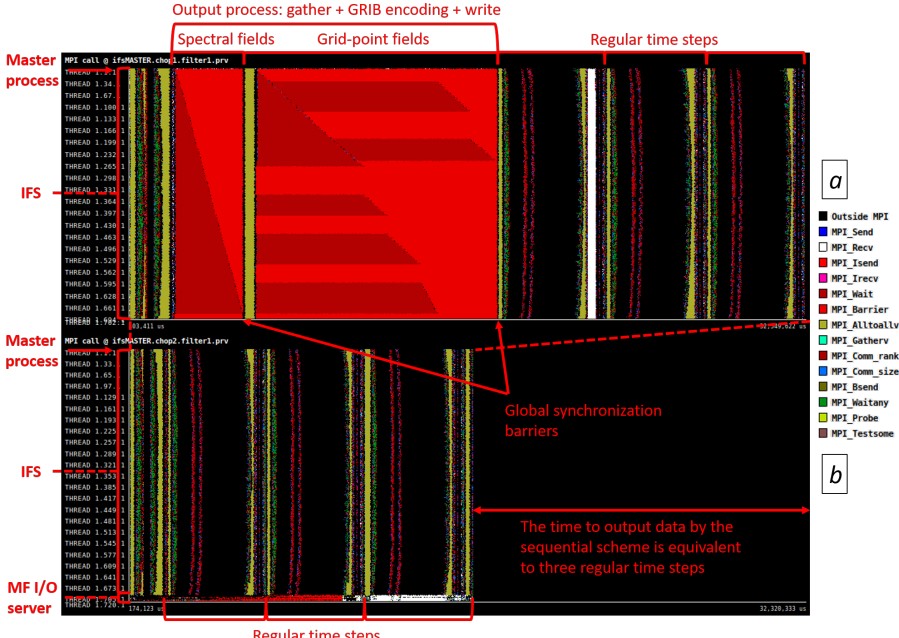

**Figure 2.** Trace *a* illustrates the serialisation caused by the sequential output scheme, whereas trace *b* illustrates that IFS with the internal parallel I/O server runs efficiently with no observable delays.

## 3.2 Profiling of communication and I/O

In order to corroborate the described performance of the previous two schemes, two IFS CY43R3 forecasts are profiled using Extrae and Paraver tracing tools (see Appendix B) on the ECMWF Cray HPCF (see Appendix C). The trace of IFS with the sequential output scheme, as expected, reveals a huge serialisation in the gather of data. Trace *a* of Fig. 2 shows the master

5 process (first process in the trace) receives a point-to-point message from each one of the rest of IFS MPI processes. In addition, between received messages, the master process performs some computation corresponding to the GRIB encoding. Meanwhile, the rest of model processes are blocked by a global synchronisation barrier. The trace shows the time devoted to perform the gather, encoding and writing is equivalent to the time needed to run three regular model timesteps. Note there are two regions of gathers in the output area: the first one, which is the smaller, corresponds to spectral fields, and the second one corresponds

10 to 3D gridpoint fields. Although this output scheme works acceptably when running low resolutions, or with much reduced model output, it is clearly insufficient at higher resolutions.

Trace *b* of Fig. 2 also indicates that the IFS with the parallel I/O server enabled runs smoothly and efficiently, without experiencing observable delays and achieves an adequate performance. This is possible due to the use of non-blocking asynchronous communication (*MPI_Isend*) and a proper data distribution among I/O servers.

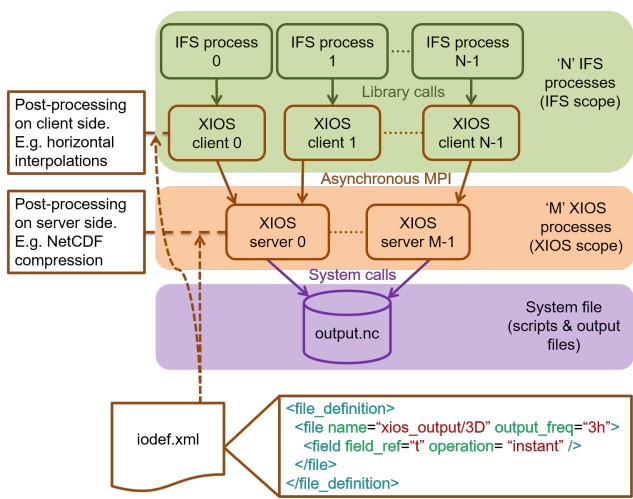

**Figure 3.** Scheme of the IFS CY43R3 and XIOS 2.0 integration. It overviews how the different parts are interconnected.

## 4  Integration of XIOS 2.0 into IFS CY43R3

### 4.1  Framework design

Figure 3 shows the integration scheme and illustrates how the different parts of IFS CY43R3 and XIOS 2.0 are interconnected. The IFS processes, shown in green, execute the client side of XIOS through its library Application Programming Interface

(API). They send data using non-blocking MPI communication to XIOS server processes, colored orange. Finally, these servers send data to the storage system (in purple) through the NetCDF library and parallel I/O middleware. Both XIOS clients and servers are configured with XML files. Note that post-processing is performed on both clients and servers. The type of post-processing determines whether some operations are performed on client side, such as horizontal interpolations, and some other on server side, such as NetCDF compression.

In order to initialise XIOS correctly, a series of steps are needed. The MPI configuration in XIOS must first be initialised, noting that IFS and XIOS binaries are run on Multiple-Program Multiple-Data (MPMD) mode thus sharing the *MPI_COMM_WORLD* communicator. Then it is essential to setup the XIOS *context*, which includes the calendar and the grid geometry. The former informs XIOS how timestep numbering translates to time stamps and the latter communicates the spatial (and parallel) distribution of the arrays that are sent from IFS, including a horizontal domain and a vertical axis.

During the execution, IFS informs XIOS about the current timestep and does the following steps:

1. Running FullPos to perform vertical interpolations, spectral smoothing and other preliminary post-processing of internal IFS fields.

2. Transforming spectral fields to gridpoint space using IFS's internal TRANS package (ECMWF, 2017), which yields extra inter-node communication overhead.





3. Copying the fields to longitude, latitude and vertical levels arrays suitable for XIOS. IFS uses a cache-friendly tiled data structure for the gridpoint fields to optimise memory accesses during the physics parameterisation routines. This data needs to be reshuffled before sending the fields to the XIOS output servers, an operation which is completely local w.r.t. the MPI decomposition of the domain.

4. Sending the fields to XIOS servers, which aggregate the data and either store the data in memory buffers or write them to disk.

Note that step 2-4 are not carried out when XIOS signals that a field is not requested at a certain timestep. Once the simulation finishes, XIOS must be finalised, which internally finalises MPI as well.

Also note that in our implementation, spectral fields are transformed onto the native reduced Gaussian grid of IFS. This
means that a subsequent interpolation to a regular lat-lon grid by XIOS introduces interpolation errors w.r.t. a direct transform. Furthermore, we have adopted all conventions from the ECMWF model: fluxes are being accumulated in the XIOS output (unless specified otherwise in the IFS configuration), the direction of vertical model level numbering is downward and we do not apply any unit conversions or arithmetic transformations. Users requiring different behaviour may therefore need to configure XIOS and IFS to produce the fields they need.

## 4.2 Implemented optimisations

This section describes the computational behaviour of the implementation of XIOS 2.0 in IFS CY43R3 to detect performance issues and possible optimisations. As we will show below (see Sect. 5), the performance of the IFS CY43R3 and XIOS 2.0 integration is adequate in terms of computational efficiency, but the XIOS 2.0 performance depends on the machine (especially the file system) and on the output size of the NetCDF files.

To anticipate potential performance bottlenecks, two different optimisations are included in the integration which are switchable at runtime through the XML input configuration files of XIOS. Although these optimisations might not always be useful, the user can test them without recompiling the code in case they offer a performance advantage.

IFS CY43R3 internally uses double (8 byte) precision numerics, but for many output fields, single precision representation in the NetCDF file suffices. Thus, the first optimisation consists of sending data from IFS processes to XIOS servers in single
precision instead of double precision to considerably reduce the transferred data volumes through the cluster network.

The second optimisation aims to take advantage of overlapping IFS computation with communication from IFS to XIOS servers. Although data is sent to XIOS using asynchronous communications, the transfer is initiated at the beginning of the timestep, whereas IFS performs synchronous internal communications. These IFS communications may be stalled because nodes and network are already occupied by data transfers involving the I/O servers. We include an option to delay the I/O data
transfers until the physics tendencies computation, which accounts for a large portion of the compute cost and is free of MPI communication. The two traces in Fig. 4 illustrate this. At the beginning of the timestep data should be output, but instead it is buffered. Then in the physics computation region of the same timestep, buffers are flushed and data is sent to the XIOS servers.

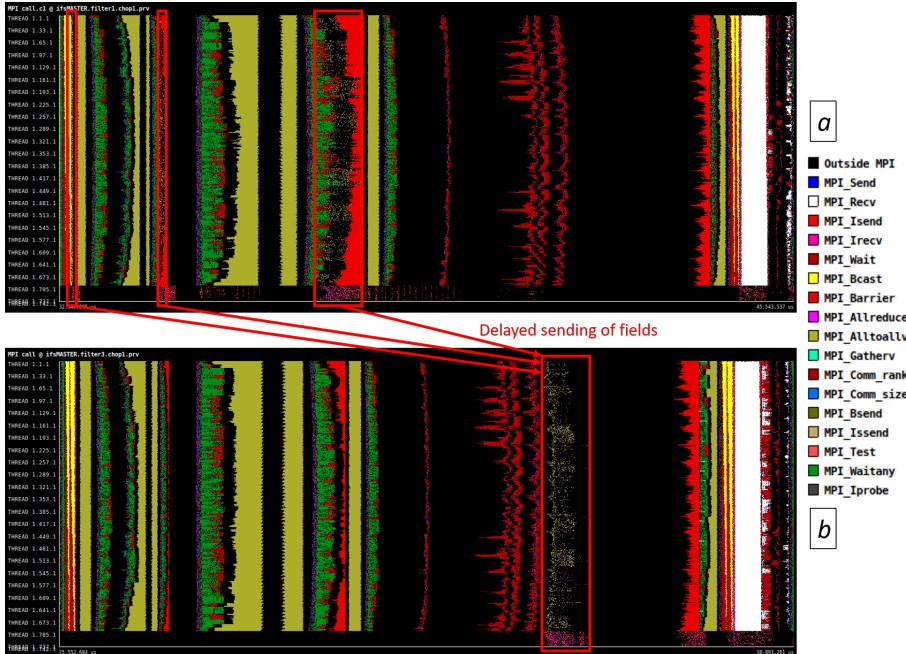

**Figure 4.** Trace *a* illustrates where fields are sent if the computation and communication overlapping optimisation is disabled, whereas trace *b* illustrates the delayed sending of the fields until the physics tendencies computation if the optimisation is enabled.

Note that data is buffered at three different points corresponding to non-lagged surface fields, spectral fields and 3D gridpoint fields.

## 5  Performance evaluation

### 5.1  Model setup

5  The setup of all tests is based on three different configurations of the IFS CY43R3 as run on the ECMWF Cray HPCF (see Appendix C). For each setup, different cases are compared: sequential output, IFS I/O server, XIOS server and no output, which serves as a reference to measure the overhead of the output schemes.

Table 1 shows a summary of the IFS CY43R3 parameters for the three different Model Intercomparison Project (MIP) configurations (Gates et al., 1999) that use a cubic octahedral Gaussian grid. AMIP and HighResMIP tests are based on analogous experiments of the CMIP6 project, without having inserted the appropriate CMIP6 metadata. The third test as its name suggests is a theoretical MIP experiment aimed to stress both the IFS-XIOS interface code and XIOS 2.0 itself to be as close as possible to future high-resolution MIP configurations (matching the current ECMWF operational horizontal resolution). Depending on the configuration, we use a different amount of computational resources to strike a good balance between performance and efficiency. Note that we also use different resource allocations when comparing the output schemes,





**Table 1.** Summary of the IFS CY43R3 configurations as run on the ECMWF Cray HPCF.

|  | AMIP | HighResMIP | Theoretical MIP |
|---|---|---|---|
| Horizontal resolution | 255 (39 km) | 511 (23 km) | 1279 (9 km) |
| Vertical resolution | 91 levels | 91 levels | 137 levels |
| Forecast length | 10 days | 10 days | 5 days |
| Timestep | 2700 seconds | 900 seconds | 600 seconds |
| Output frequency | 3 and 6 hours | 3 and 6 hours | 1 hour |
| IFS MPI processes | 22 (2 nodes) | 56 (5 nodes) | 702 (59 nodes) |
| IFS processes per node | 12 | 12 | 12 |
| OpenMP threads per process | 6 | 6 | 6 |
| IFS I/O server MPI processes | 2 (1 node) | 4 (1 node) | 30 (3 nodes) |
| XIOS MPI processes | 2 (1 node) | 4 (2 nodes) | 40 (20 nodes) |
| Hyper-threading | Yes | Yes | Yes |
| Output size (GRIB) | 20 GB | 77 GB | 2.4 TB |
| Output size (NetCDF) | 48 GB | 206 GB | 9.9 TB |

as it makes no sense to allocate dedicated resources for the sequential scheme. The IFS I/O server and XIOS require different amounts of resources (see Sect. 5.2) due to the output volume being considerably smaller for the GRIB output than the XIOS-produced NetCDF files. This is a result of the intrinsic (lossy) compression in GRIB, where most horizontal field slices are scaled and translated to be encoded in 16-bit numbers. Conversely, we have not configured XIOS to use NetCDF compression, so every value is represented by a 32-bit float. We will elaborate on this in Sect. 6.

We note that a tuning of the compilation, the job scheduling and configuration of the hardware platform may significantly increase the performance of the I/O scheme. We have used the `-O3` optimisation flag to compile XIOS, which yields a notable improvement of the execution time. Secondly, we carefully configure the process affinity of the two parallel binaries which have such different computational patterns. Although both IFS CY43R3 and XIOS 2.0 are run in Multiple-Program Multiple-Data (MPMD) mode, independent affinity for each component is crucial: we tune the number of processes, number of processes per node and number of processes per Non-Uniform Memory Access (NUMA) socket within a node for maximal performance. Finally, we configure the Lustre parallel file system at the ECMWF Cray HPCF. By default, Lustre only uses one Object Storage Target (OST) device to store a file. However, it is possible to split the file into chunks that are stored in different OSTs. This is known as striping, and it is applied to the XIOS output NetCDF files to improve the performance, especially for very large files.

Section 5.2 shows the optimal number of XIOS servers in relation to the output size. Subsequently, Sect. 5.3 performs a comparison between all the output schemes previously mentioned. Section 5.4 performs the same type of comparison, but





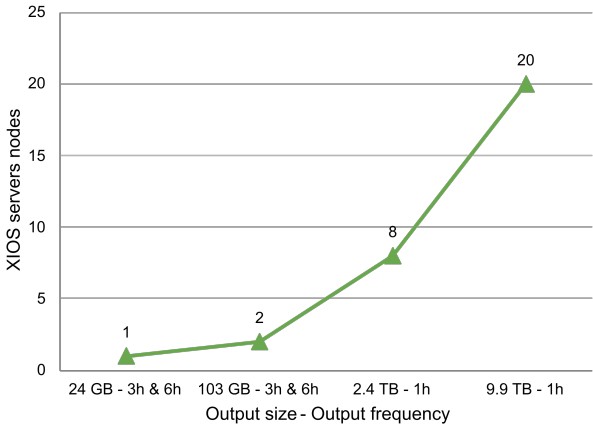

**Figure 5.** Number of nodes needed by XIOS 2.0 servers in relation to the output size and writing frequency.

adding the time needed to convert GRIB to NetCDF files to simulate a production Earth system model that requires delivery of products to end-users within a specified time. The results of these sections are obtained from the average of three runs.

## 5.2 Resources usage of XIOS 2.0 servers

XIOS 2.0 servers consume a considerable amount of memory, for example whilst accumulating data before flushing to NetCDF
5  output. As a consequence, a minimum number of exclusive cluster nodes may need to be allocated to provide sufficient memory. This number depends on the model resolution, the number of output variables, the output frequency and has to be determined by trial and error. The more XIOS servers are working at the same time, the more fields can be processed in parallel, avoiding a potential bottleneck during the output process. Figure 5 uses four different output sizes: the first configuration uses a resolution of ~39 km on a cubic octahedral grid with 91 vertical levels (Tco255L91); the second configuration uses a resolution of ~23
10  km on a cubic octahedral grid with 91 vertical levels (Tco511L91); and the last two configurations use a resolution of ~9 km on a cubic octahedral grid with 137 vertical levels (Tco1279L137) with different sets of output fields. We observe that the required number of output nodes grows significantly with the output size, especially for 2.4 and 9.9 TB cases that need at least 8 and 20 nodes respectively. In these configurations we place only a single XIOS MPI task per output node, since there is scarce computational demand on the post-processing server side.

15  ## 5.3 Output schemes comparison

This section compares the computational performance of the different output schemes described in Sect. 3. We measure the I/O overhead of the scheme by comparing the model wall-clock time with the 'no output' scheme (Fig. 6), which does not write any actual output, but does perform vertical interpolations with FullPos of the requested output fields. We also include a no output version, 'no output+spectral', that performs the spectral transforms needed to send dynamical fields in gridpoint



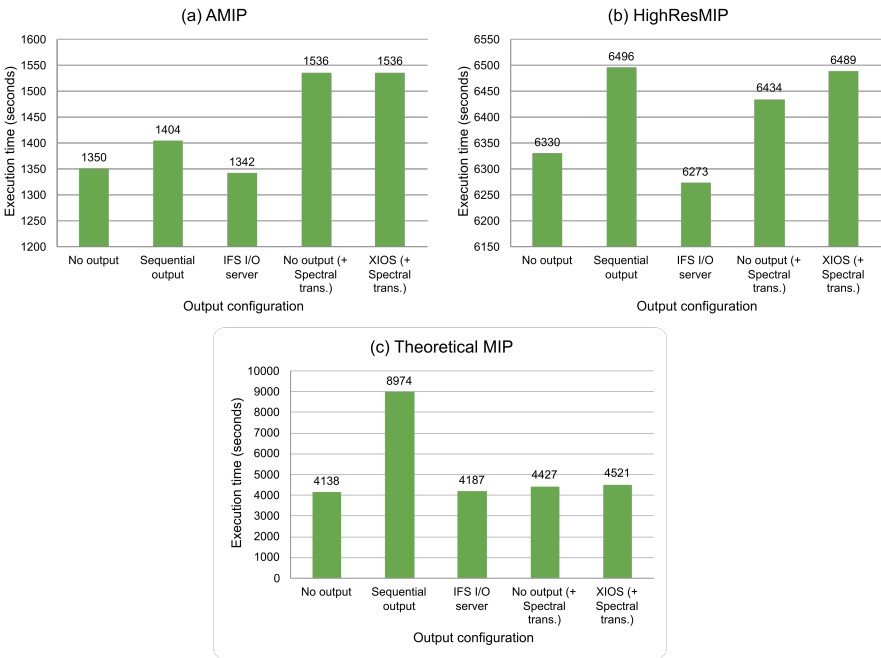

**Figure 6.** Comparison of the execution time of different output schemes for the three configurations evaluated.

space to XIOS. Whereas sequential output and IFS I/O server have to be compared with the 'no output' case, XIOS has to be compared with the 'no output+spectral' case.

Figure 6 contains three different plots corresponding to the three configurations described above. We first notice that the sequential output scheme does not scale at all with the output volume and its I/O overhead increases model wall-clock time up
to 54%. On the contrary, the IFS I/O server has a low I/O overhead in all configurations, even for the Theoretical MIP one (less than 1.2%). It it important to remark that for AMIP and HighResMIP configurations the execution time of the IFS I/O server is smaller than the no output case, which might be attributed to the variability of executions. The I/O overhead of XIOS is also small, which is less than 2.1% for the Theoretical MIP case. The I/O overhead of XIOS is only slightly higher than the IFS I/O server, which may well be a consequence of the much larger output size of the NetCDF output (9.9 and 2.4 TB respectively).
The optimisation efforts described in Sect. 4.2 were tested in these experiments but were found to have no significant speedup. Nevertheless we have kept the options in the code, since other configurations may benefit from them.

## 5.4 Comparison including post-processing

This test is the same as the previous Sect. 5.3, but adds the cost of transforming spectral fields as well as converting GRIB files to NetCDF files. This cost is only added to the sequential output and IFS I/O server schemes as they write data in GRIB
format. This test is useful to know the potential benefit of using XIOS to avoid the costly post-processing performed in climate simulations.





Figure 7 compares the three output schemes, where the post-processing script consists of the following steps:

1. Filter the GRIB files produced by IFS according to level type, which is necessary for Climate Data Operator (CDO) to be able to read them. In this step, we also concatenate the single-timestep files.

2. For the gridpoint fields, remap to a regular grid, and for the spectral fields, do a spectral transform using the `sp2gpl` tool in CDO.

3. Write all data back in NetCDF4 uncompressed format.

The last two steps are carried out concurrently for the co-ordinate and spectral space variables and vertical axis types. The post-processing steps are made sequentially for all timesteps on a single node. We consider this script to represent the minimal post-processing workflow for climate applications of IFS, such as EC-Earth production runs. Almost all monitoring and post-processing scripts and tooling for additional diagnostics perform these steps in one way or another.

Post-processing of the AMIP configuration takes about 523 seconds, HighResMIP configuration about 4497 seconds and post-processing only the last output timestep of the Theoretical MIP configuration takes about 1845 seconds. The estimation to post-process all 120 output timesteps of this last configuration would take about 221,400 seconds (2.56 days), unless 120 nodes where concurrently allocated to post-process all 120 output timesteps.

After including this post-processing penalty, the true benefit of the XIOS integration becomes apparent. As shown in Fig. 7, the XIOS configuration outperforms the IFS I/O server by 21.4%, 66% and 33.4% for AMIP, HighResMIP and Theoretical MIP configuration respectively. For the sequential scheme the speedup is even larger, being 25.5%, 69.4% and 139.3% respectively. Note we only included post-processing of the last timestep output for the Theoretical MIP configuration.

## 6 Data validation

To compare the XIOS-produced NetCDF output with the original GRIB files, we configured the XIOS I/O server to produce all variables on the original reduced Gaussian grid for the Tco255L91 resolution AMIP-benchmark run. The output data in the NetCDF files is represented by 32-bit floating point numbers, whereas the original GRIB fields are usually encoded into 16-bit numbers, translated and scaled to span the interval defined by the minimum and maximum of the field. The scaling and offset is defined for every record separately, and is therefore nonuniform across time steps and model levels.

The resolution of a GRIB record with field values $f$ and encoding bits $B$ is therefore given by

$$\epsilon = \frac{\max(f) - \min(f)}{2^B} \tag{1}$$

This resolution dominates the error on the difference between the original output and the XIOS-produced data, since it is orders of magnitude larger than the internal floating-point error for the typical value $B = 16$ used at ECMWF. In Fig. 8 we depict the vertical profile of the error in specific humidity, showing that the maximal difference between the two output methods is compatible with zero within error bounds.



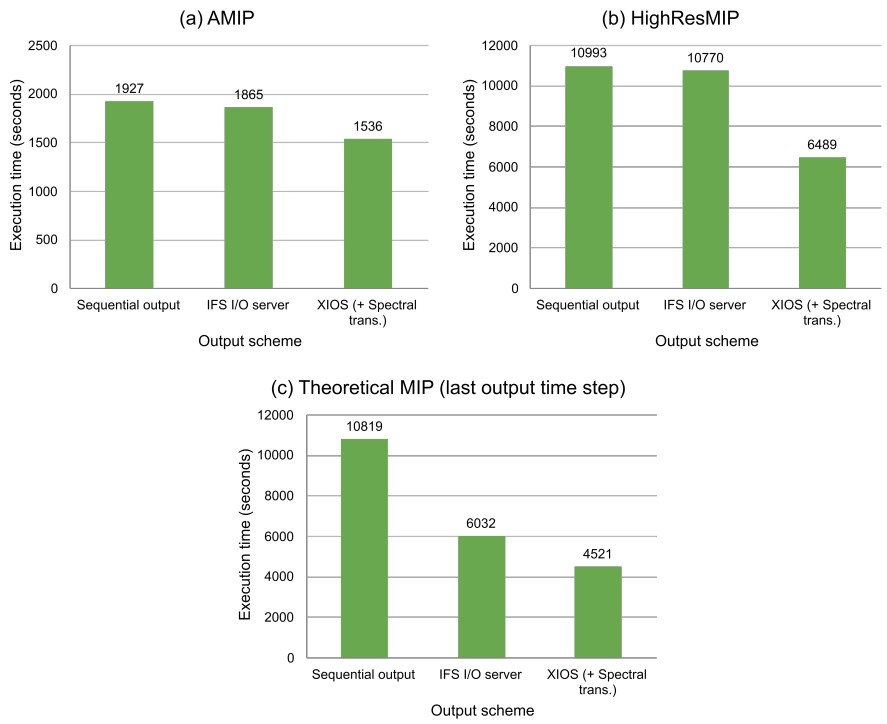

**Figure 7.** Comparison of the execution time of different output schemes for the three configurations evaluated: both sequential output and IFS I/O server have the additional cost of transforming spectral fields and converting GRIB files to NetCDF files.

The same conclusion is drawn for a selection of surface fields in Fig. 9, where the relative differences and errors are depicted by dividing with the lower bound of the GRIB records. Finally, no spatial patterns in the error along the horizontal directions have been observed throughout the benchmark output, nor have any irregularities regarding the transformed spectral fields other than expected interpolation errors.

## 7 Summary and conclusions

In this paper we have integrated XIOS 2.0 into IFS CY43R3 with a twofold objective: provide IFS with a more flexible output tool to comply with widely accepted standards in the climate modelling community; and replace the obsolete sequential I/O scheme that would otherwise incur a heavy performance penalty for high-resolution model configurations in Earth system models derived from IFS. The use of XIOS provides the following advantages:

1. Improve scalability of IFS derivatives as XIOS is capable of writing in parallel.

2. Produce NetCDF files by IFS derivatives and if configured appropriately, with user-provided metadata to be CMIP-compliant. XIOS is also capable of post-processing the IFS fields and computing additional diagnostics inline.

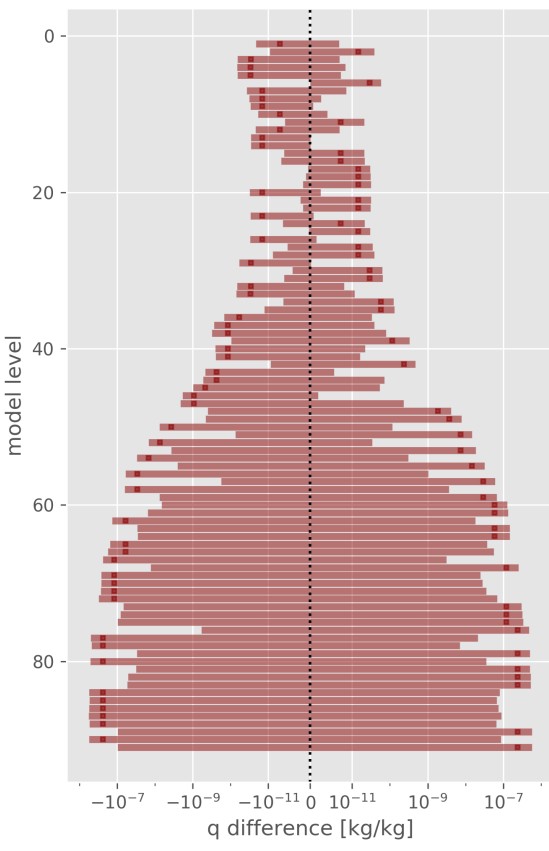

**Figure 8.** Maximal absolute errors (over gridpoints and timesteps) for specific humidity, for each model level. The error bars denote the error on the GRIB values, obtained by evaluating Eq. (1).

3. Provide a more integrated output configuration strategy for coupled models which employ XIOS already. Also, derived fields combining output from multiple sub-models will be feasible.

4. Increase the impact of the XIOS initiative throughout the European climate and weather modelling landscape to secure a high-performance I/O strategy.

5   In order to achieve these objectives, we presented a development effort which performs a series of steps when model output is required: run FullPos to perform vertical interpolations; transform spectral fields to gridpoint space; reshuffle IFS data arrays from a cache-friendly tiled data structure to longitude, latitude and vertical levels arrays suitable for XIOS; and send fields to XIOS servers.

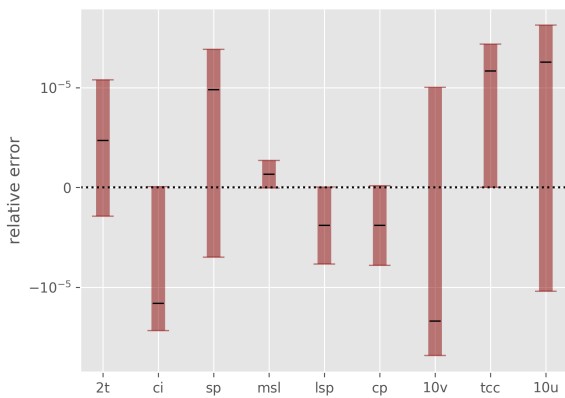

**Figure 9.** Maximal relative errors (over gridpoints and timesteps) for a few surface fields. The error bars denote the error on the GRIB values, obtained by evaluating Eq. (1).

The performance evaluation shows the XIOS server outperforms the sequential I/O scheme. While the latter does not scale at all as its overhead is up to 54%, XIOS has a low overhead of less than 2.1%. However, the overhead of the IFS I/O server is slightly better (less than 1.2%), but this might be due to the difference in size of the data (9.9 and 2.4 TB for NetCDF and GRIB files respectively).

When we also take into account the post-processing task to compare performance results, the benefit of using XIOS is greater. The post-processing of sequential output and IFS I/O server schemes consists of converting GRIB files to NetCDF files as well as transforming spectral fields. These two operations are expensive in terms of wall-clock time, especially when the output volume is large. For example, the time devoted to post-process AMIP and HighResMIP output configurations is about 523 and 4497 seconds, increasing the IFS I/O server time by 21.4% and 66% respectively, and increasing the sequential output

scheme time by 25.5% and 69.4% respectively.

    We have also ported the integration to OpenIFS 43R3V1. This aims to shorten the critical path of Earth system models that use an IFS CY43R3 derivative or OpenIFS 43R3V1 as their atmospheric component by optimising the I/O scheme and providing inline post-processing. It is planned that OpenIFS 43R3V1 will replace IFS CY36R4 in EC-Earth 4, with a foreseeable saving of thousands of core hours and storage space.

**Appendix A: Description of the components**

**A1   IFS and OpenIFS**

The Integrated Forecasting System (IFS) (Barros et al., 1995) (ECMWF, b) is an operational global meteorological forecasting model and data assimilation system developed and maintained by ECMWF. It is a spectral model that discretises the Euler



equations of motion, resolving flow features to approximately 4-6 grid-cells at the nominal resolution. The subgrid-scale features and unresolved processes are described by atmospheric physics parametrisations. There are many different unresolved physical processes in the atmosphere, such as radiation, clouds and subgrid turbulent motions.

The dynamical core of IFS is hydrostatic, two-time-level, semi-implicit, semi-Lagrangian and applies spectral transforms
between gridpoint space (where the physical parametrisations and advection are calculated) and spectral space. In the vertical the model is discretised using a finite-element scheme. A cubic octahedral reduced Gaussian grid is used in the horizontal.

OpenIFS is derived from IFS and designed to be run on systems external to ECMWF. It has the same forecast capability of IFS where the data assimilation or observation handling functionality has been removed. It is licensed software provided free to research and educational institutes.

## A2   XIOS

The XML Input/Output Server (XIOS) (Joussaume et al., 2012) (Meurdesoif et al., 2016) (Hanke et al., 2013) (Maisonnave et al., 2017) (http://forge.ipsl.jussieu.fr/ioserver) is an asynchronous MPI parallel I/O server used by Earth system models to avoid contention in their I/O. It focuses on offering high performance to achieve very high scalability with support for high-resolution output. XIOS is developed by the Institute Pierre Simon Laplace (IPSL). It has the following features:

– Usability in the definition and management of the I/O with a user-friendly Extensible Markup Language (XML) configuration file.

– Avoid the I/O performance issue with dedicated parallel and asynchronous servers.

– Post-processing of fields can be performed inline using an internal parallel workflow and dataflow.

XIOS is especially targeted to Earth system models with these characteristics: coupled models, long simulations, a lot of
model data generated and contributed to the CMIP project. They are inherent in climate models, such as EC-Earth.

In Fig. A1 there is an overview of the schematic architecture used in XIOS. Each one of the model processes runs its own XIOS client using the XIOS API. This is part of the client side, i.e., it is run on the model processes. Then, XIOS clients communicate data to XIOS servers using asynchronous MPI messages. They are run on independent nodes with regard to the nodes running the model. This is the server side, which uses its own MPI communicator to perform inline post-processing over
the received data. After that, XIOS servers can write post-processed data into the storage system using two different strategies: one single file or multiple files (one per XIOS server).

Furthermore, although Fig. A1 shows an XIOS configuration using the server mode (dedicated I/O processes), it is also possible to use the client mode. In this case XIOS clients are responsible of post-processing and writing data into the storage system, and will be blocking the model time stepping in doing so.





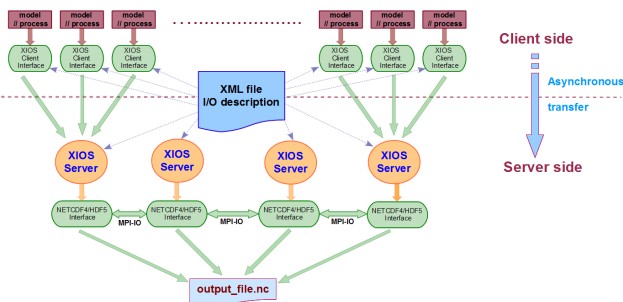

**Figure A1.** Overview of the XIOS architecture. Model processes are communicated with the XIOS servers using asynchronous MPI messages. All the framework is configured using an XML file (Reproduced from Meurdesoif et al. (2016)).

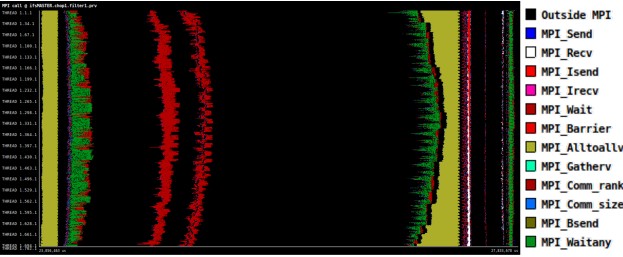

**Figure B1.** Example of a trace with MPI call events. Timeline is on the *x* axis and MPI processes on the *y* axis.

## Appendix B: BSC profiling tools

Tracing is the process of recording event-based performance data along the execution of a program. Using a viewer it is possible to see the behaviour of the application in the machine used, focusing on hardware counters, communication patterns or memory hierarchy. The tools used to trace the model are Extrae and Paraver, which are open-source and developed by the BSC Performance Tools group (https://tools.bsc.es):

- Extrae is a package used to instrument the code automatically and/or manually through its API. It generates Paraver trace-files with hardware counters, MPI messages and other information for a post-mortem analysis.

- Paraver is a trace browser that can show a global visual qualitative perspective of the application behaviour for later focus on the bottlenecks with a detailed quantitative analysis. The tool allows to create views with any parameter recorded and points to a region of a code, making process of modification easier.

Figure B1 shows an example of a trace, which has the timeline on the *x* axis and the MPI processes on the *y* axis. Along the timeline, some (or many) events happen, which can be related to MPI calls, cache misses, MIPS and many other performance metrics. The trace of the Fig. B1 shows MPI call events, where each color represents an MPI function. Note that the first color, black, it is actually not an MPI function, but it represents computation (outside MPI).



## Appendix C: HPC platform and compilation environment

ECMWF's High Performance Computing Facility (HPCF) (ECMWF, a) is a Cray system that has two identical Cray XC40 clusters. It has a peak performance of 8499 teraflops.

Each Cray XC40 cluster has 3610 compute nodes running the Cray CLE 5.2 UP04 operating system. Nodes are made of 128
GB of memory and two Intel E5-2695v4 "Broadwell" processors, each with 18 cores. It is possible to activate hyper-threading, offering a total of 72 threads.

Cores have three levels of cache available, L1, L2 and L3, with 64 KiB, 256 KiB and 45 MiB (shared) of memory respectively. They operate at a clock frequency of 2.1 GHz.

The cluster uses the Aries™ Interconnect network technology, which implements a "dragonfly" topology. In addition, Lustre
is used as the parallel file system.

The Cray Developer Toolkit (CDT) version 16.04 and the Cray Compiling Environment (CCE) version 8.4.6 were used to compile both IFS CY43R3 and XIOS 2.0. They were built with Cray MPICH 7.3.3.

*Code and data availability.* The source code of the integration is available on Zenodo (https://doi.org/10.5281/zenodo.4905832) and on GitLab (https://earth.bsc.es/gitlab/xyepes/ifs-cy43r3-and-xios-2.0-integration). GRIB and NetCDF dataset files of the AMIP configuration
can be used to check the correctness at: https://doi.org/10.5281/zenodo.4473008. The IFS source code is available subject to a license agreement with ECMWF. ECMWF member-state weather services and their approved partners will be granted access. The IFS code without modules for data assimilation is also available for educational and academic purposes as part of the OpenIFS project (https://confluence. ecmwf.int/display/OIFS/OpenIFS+Home), which includes the integration with XIOS. The XIOS source code is available on Zenodo (https://doi.org/10.5281/zenodo.4905653) and on the official repository (http://forge.ipsl.jussieu.fr/ioserver). The validation scripts are available on
Zenodo (https://doi.org/10.5281/zenodo.4906175) and on GitHub (https://github.com/goord/xios-grib-compare).

*Author contributions.* XY-A lead the paper, all co-authors contributed to writing sections and reviewing the paper. XY-A and GvdO developed the XIOS interface code. XY-A carried out the performance tests on the ECMWF system. GvdO performed the diagnostics study. GC provided support for implementation in OpenIFS. MA helped in conceptual design as well as reviewed the performance evaluation tests and results.

*Competing interests.* The authors declare that they have no conflict of interest.

*Acknowledgements.* We would like to thank to Stéphane Senesi, Yann Meurdesoif, Etienne Tourigny, Xavier Abellan, Daniel Jiménez-González, Mats Hamrud, Ryad El Khatib, Kim Serradell, Pierre-Antoine Bretonniére, Nils Wedi, Kristian Mogensen, Francisco J. Doblas-Reyes, Philippe Le Sager, Uwe Fladrich, Klaus Wyser, Rein Haarsma, Camiel Severijns, Miguel Castrillo and Oriol Tintó-Prims for helpful



discussions. We also express our gratitude to ECMWF for providing computational resources to carry out the development of the IFS-XIOS integration and run many different experiments. This work has been funded by ESiWACE2 (GA 823988) and PRIMAVERA (GA 641727) EU Commission projects.



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
