# Peer review of "Evaluation and optimisation of the I/O scalability for the next generation of Earth system models: IFS CY43R3 and XIOS 2.0 integration as a case study"

_Geoscientific Model Development, 2021_

## Author Comment (AC1)

**Reply to RC1: Anonymous Referee #1**

This is an excellent paper reporting on the experience of implementing XIOS, an IO server into the ECMWF intetgrated forecast system. The paper has a good introduction and motivation with well presented results and discussion. I can't remember the last time I read something on this sort of topic which was as easy to consume. I find myself with little to contribute in terms of suggested modfications!

I recommend publishing as is, but recommend addressing a few minor points during the final submission process. That said the authors and editors might consider whether section 6 and the associated figures are really necessary? It's not really about choices around XIOS, but about the difference between GRIB and NetCDF ... it's a different topic …

Reply: We are grateful for the constructive and insightful comments and we have addressed all the specific issues raised by the reviewer. The original reviewer comments are colored in blue.

We believe a proper validation of the model output is critical when adopting a new I/O infrastructure. When comparing the XIOS output -restricted to NetCDF format- with the traditional scheme -restricted to GRIB format- one unfortunately cannot escape the details of the file formats, different compressions, and its implications for the validation procedure.

My list of minor points:

- obsolete not obsolescent (in the abstract, it is correct in the text)

Reply: This is now corrected.

- I would recommmend adding the word "existing" before "IFS I/O server" at the end of page 1.

Reply: We prefer to add the word "operational" before "IFS I/O server" as described in the second paragraph of page 1 to keep the descriptions consistent.

- line 5 page 2. I would discuss increasing spatial resolution, not spatio-temporal resolution (even if the timestep has changed).

Reply: This is now corrected.

- For the caption of figure 2, I think it would be helpful to expand a little bit more about the meaning of the axes in this trace. While it is easy to infer, it takes a few minutes for those who are not familiar with the output. I realise this is discussed further in B1, but frankly most readers might not get to the appendix!

Reply: We have added a brief explanation about the meaning of the trace axes.

- the first sentence of the last para of page 9 is hard to construe given the previous para talks about "the three tests" and then this para talks about "the three MIP configurations). After two or three reads and a look at the table I understood it, but it could be cleaner.

Reply: We have fixed this by only using "configuration" instead of indistinctly using both "test" and "configuration".

---

## Author Comment (AC2)

**Reply to RC2: Jim Edwards**

Overall an excellent paper.   On the issue of scientific reproducibility, that the IFS parallel IO scheme is not publically available makes this aspect of the paper difficult or impossible to reproduce.

Reply: We are grateful for the constructive and insightful comments and we have addressed all the specific issues raised by the reviewer. The original reviewer comments are colored in blue.

It is correct that IFS with the I/O server code is not "publicly" available, but the code is available as described in the "Code and data availability" section. It is also important to mention that in practise exact reproducibility would be impossible to achieve given the dependence on the HPC hardware, software stack and underlying file system.

I only have a couple of suggested corrections:

Page 13, line 14: "where" should be "were"

Reply: This is now corrected.

Page 17, line 28: "of" should be "for"

Reply: This is now corrected.

Page 17, line 29: " will be blocking the model time stepping in doing so" should be "will block progress of the model time step in doing so"

Reply: This is now corrected.

I would also like to point out newer reference documents for the parallelIO library: https://github.com/NCAR/ParallelIO#references

Reply: We have changed the old reference for these two newer ones:
- Hartnett, E., Edwards, J., "THE PARALLELIO (PIO) C/FORTRAN LIBRARIES FOR SCALABLE HPC PERFORMANCE", 37th Conference on Environmental Information Processing Technologies, American Meteorological Society Annual Meeting, January, 2021. Retrieved on Feb 3, 2021, from [https://www.researchgate.net/publication/348169990_THE_PARALLELIO_PIO_CFORTRAN_LIBRARIES_FOR_SCALABLE_HPC_PERFORMANCE].
- Hartnett, E., Edwards, J., "POSTER: THE PARALLELIO (PIO) C/FORTRAN LIBRARIES FOR SCALABLE HPC PERFORMANCE", 37th Conference on Environmental Information Processing Technologies, American Meteorological Society Annual Meeting, January, 2021. Retrieved on Feb 3, 2021, from [https://www.researchgate.net/publication/348170136_THE_PARALLELIO_PIO_CFORTRAN_LIBRARIES_FOR_SCALABLE_HPC_PERFORMANCE].